# HYSYNTH: Context-Free LLM Approximation for Guiding Program Synthesis

**Shraddha Barke**
UC San Diego
San Diego, USA
sbarke@ucsd.edu

**Emmanuel Anaya Gonzalez**
UC San Diego
San Diego, USA
fanayagonzalez@ucsd.edu

**Saketh Ram Kasibatla**
UC San Diego
San Diego, USA
skasibatla@ucsd.edu

**Taylor Berg-Kirkpatrick**
UC San Diego
San Diego, USA
tbergkirkpatrick@ucsd.edu

**Nadia Polikarpova**
UC San Diego
San Diego, USA
npolikarpova@ucsd.edu

## Abstract

Many structured prediction and reasoning tasks can be framed as program synthesis problems, where the goal is to generate a program in a *domain-specific language* (DSL) that transforms input data into the desired output. Unfortunately, purely neural approaches, such as large language models (LLMs), often fail to produce fully correct programs in unfamiliar DSLs, while purely symbolic methods based on combinatorial search scale poorly to complex problems. Motivated by these limitations, we introduce a hybrid approach, where LLM completions for a given task are used to learn a task-specific, context-free surrogate model, which is then used to guide program synthesis. We evaluate this hybrid approach on three domains, and show that it outperforms both unguided search and direct sampling from LLMs, as well as existing program synthesizers.

## 1 Introduction

Large language models (LLMs) demonstrate impressive capabilities in various domains, but they continue to struggle with tasks that require precision—e.g. structured prediction, reasoning, counting, or data transformation—when direct task examples are not prevalent in their training data [8, 12, 23, 31, 38, 40, 45]. As one example, consider the *Abstraction and Reasoning Corpus* (ARC) [14], which was designed as a benchmark for human-like structured reasoning. ARC tasks are grid-based puzzles, such as one depicted in Fig. 1a. This puzzle consists of three training examples, which are pairs of input and output grids; the goal is to infer the transformation that maps the input to the output, and then apply this transformation to the test grid. The ARC benchmark's emphasis on generalization and few-shot learning has rendered it challenging to solve with purely machine learning techniques: state-of-the-art generative models like GPT-4 hardly solve more than 10% of the tasks in the dataset when asked to predict the test output, even with the help of advanced prompting techniques [25].

In fact, the leading entries in the ARC Kaggle competition [1] tackle this task using *Programming-by-Example* (PBE): instead of predicting the output directly, they search for a program that captures the transformation occurring in the input-output examples. For example, the transformation in Fig. 1a might be represented as the following program:

$$\textbf{if } \texttt{color\_of}(\textbf{self}) = \texttt{GREY} \wedge \texttt{is\_neighbor}(\textbf{self}, \texttt{other}) \wedge \texttt{size\_of}(\texttt{other}) = \texttt{MIN}$$
$$\textbf{then } \texttt{update\_color}(\texttt{color\_of}(\texttt{other})) \quad (1)$$

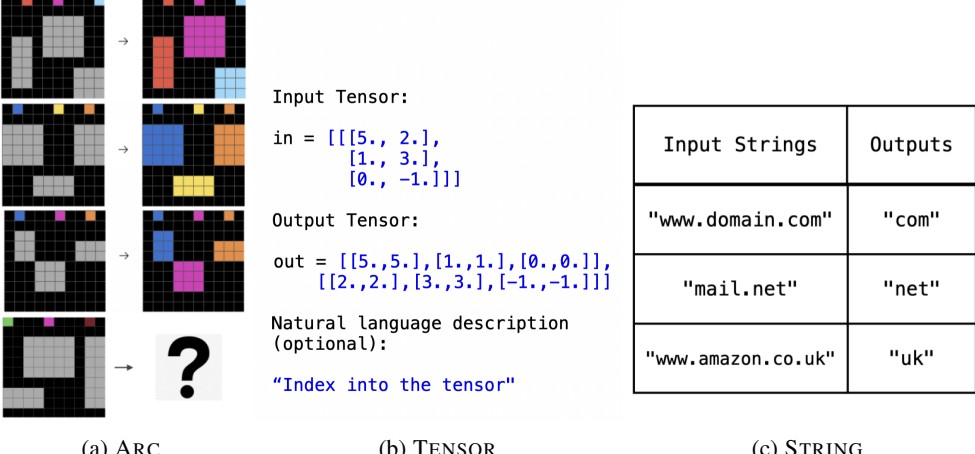

|  | (a) ARC | (b) TENSOR | (c) STRING |

Figure 1: Example problems from the three PBE domains we evaluate HYSYNTH on: grid-based puzzles (ARC), tensor manipulation (TENSOR), and string manipulation (STRING).

This particular program is written in a *domain-specific language* (DSL) inspired by the ARGA tool [44]. It consists of a single *rule* of the form **if** *filter* **then** *transform*, which is applied to each object in the grid simultaneously; if the filter holds for the focus object **self** and another object **other**, then **self** undergoes the transform. In this case, the rule says that any grey object that has a neighbor of the grid's minimum size (here, a single pixel) should be colored with the color of that neighbor.

Beyond grid puzzles, PBE is a general paradigm for structured reasoning and data transformation tasks: for example, it can help spreadsheet users with systematic string manipulation [20], and help programmers use unfamiliar APIs [17, 18, 36]; Fig. 1 shows example PBE tasks from three domains.

**Challenge: Harnessing the Power of LLMs for PBE**    How can we automatically learn programs from the input-output examples like those shown in Fig. 1? The traditional *program synthesis* approach is based on combinatorial search [2, 7, 34, 35, 39], which works well for small programs and restrictive DSLs, but becomes infeasible as the program size and the DSL complexity grow. At the other end of the spectrum, purely *neural* approaches [15, 42] use a neural model to predict the program from input-output examples; unfortunately, even state-of-art LLMs like GPT-4o [33] struggle to predict an entire program in an unfamiliar DSL: when we asked GPT-4o to generate 10 programs for the running example above, none of them were entirely correct.[1]

In the past, the limitations of both program synthesis and neural techniques have motivated a hybrid approach, where combinatorial search is *guided* by a learned probabilistic model [9, 24, 26, 32, 36, 37]. Existing hybrid techniques, however, use domain-specific models trained on datasets of similar PBE tasks, which limits their generalization to new domains. With the advent of LLMs, can we now use a single pre-trained model to guide program synthesis across a wide range of domains?

Interestingly, there is some tension in the hybrid approach between the efficiency of the search algorithm and the power of the model: a search algorithm is efficient when it *factorizes the search space* (*i.e.*, merges many search states into one), which often makes it incompatible with a powerful model that requires a lot of context to make a prediction. Specifically, one of the most widely used program synthesis techniques is *bottom-up search* [2, 11, 28, 36, 39], which is a dynamic programming algorithm, whose efficiency relies on reusing the work of constructing and evaluating subprograms in many different contexts. This essentially precludes using models with unlimited left-to-right context—like LLMs–to guide bottom-up search.

**Our Solution: Context-Free LLM Approximation**    To bridge this gap and harness the power of LLMs to guide bottom-up search, we propose to approximate the LLM's conditional output distribution *for a given task* with a context-free surrogate model. Recent work in NLP [46] has found that a Hidden Markov Model (HMM) trained to match an LLM can be used as an efficient surrogate

---

[1]A detailed analysis of GPT-4o's performance on this task is provided in Appendix A.

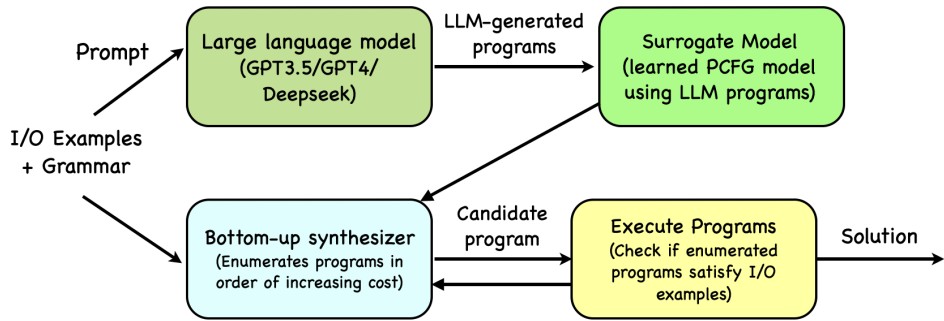

Figure 2: An overview of the hybrid program synthesis technique that uses a context-free LLM approximation. Programs generated by an LLM are used to learn a PCFG, which guides a bottom-up synthesizer to generate programs until a solution is found.

in style-controlled language generation. We extend this idea to program synthesis, replacing the HMM with a *probabilistic context-free grammar* (PCFG). The benefits of using a PCFG are twofold: (1) PCFGs are context-free, which makes them compatible with bottom-up search for PBE [11, 36], and (2) while a context-free model may make a poor approximation to an LLM's full joint, in a PBE setting it is able to reasonably approximate an LLM's conditional distribution over output programs *for a given prompt*. The overview of our approach is shown in Fig. 2.

**Evaluation**   We implemented this technique in a tool HYSYNTH[2] and evaluated it on 299 PBE tasks from three domains: ARC grid-based puzzles [14], tensor manipulation tasks from TFCODER [36], and string manipulation tasks from the SYGUS benchmark [5], which are inspired by spreadsheet use cases. Example problems from these domains are shown in Fig. 1. Our evaluation shows that HYSYNTH outperforms both unguided search and LLMs alone, solving 58% of the tasks overall, compared to 40% for unguided search and 6% for LLMs without search. Our tool also outperforms baseline program synthesizers for these domains—ARGA, TFCODER, and PROBE [11], respectively; importantly, in the TENSOR domain, the guidance from the LLM not only speeds up the search, but also frees the user from having to explicitly provide any non-standard *constants* that the solution might use, thereby significantly improving the usability of the tool.

**Contributions**   In summary, this paper makes the following contributions:

1. We propose a hybrid program synthesis approach that integrates LLMs with efficient bottom-up search via a task-specific context-free approximation.

2. We implement this approach in a tool HYSYNTH and instantiate it on three domains: grid-based puzzles (ARC), tensor manipulation (TENSOR), and string manipulation (STRING). While the latter two domains reuse off-the-shelf bottom-up synthesizers, for ARC we implement a custom synthesizer that uses a divide-and-conquer strategy [6] to leverage the structure of the rule-based DSL to further speed up the search.

3. We evaluate HYSYNTH on the three domains and show that it outperforms both the LLM alone and existing baseline synthesizers, which are not guided by LLMs.

## 2   Background

### 2.1   Programming-By-Example

Programming by Example (PBE) [21] is the task of synthesizing programs that satisfy a given set of input-output examples. To restrict the program space, the programs are typically drawn from a *domain-specific language* (DSL), which is specified by a *context-free grammar* and an *evaluation function*. This section provides a formal definition of these concepts.

**Context-Free Grammars**   A *context-free grammar* (CFG) is a quadruple $\mathcal{G} = (\mathcal{N}, \Sigma, \mathcal{S}, \mathcal{R})$, where $\mathcal{N}$ is a set of non-terminal symbols, $\Sigma$ is a set of terminal symbols, $\mathcal{S} \in \mathcal{N}$ denotes the starting

---

[2]The name stands for "HYbrid SYNTHesis" and is pronounced like the flower "hyacinth".

| | | | | |
|---|---|---|---|---|
| $Rule \rightarrow$ | **if** $Filter$ **then** $Transform$ | | $Color \rightarrow$ | color_of($Obj$) $\mid$ GREY $\mid$ RED ... |
| $Filter \rightarrow$ | $Atom \mid$ not $Atom \mid Atom \wedge Filter \mid \ldots$ | | $Size \rightarrow$ | size_of($Obj$) $\mid$ MIN $\mid$ MAX $\mid \ldots$ |
| $Atom \rightarrow$ | $Color =_c Color \mid Size =_s Size \mid \ldots$ | | $Dir \rightarrow$ | dir_of($Obj$) $\mid$ UP $\mid$ DOWN $\mid \ldots$ |
| $Transform \rightarrow$ | update_color($Color$) $\mid$ move($Dir$) $\mid \ldots$ | | $Obj \rightarrow$ | **self** $\mid$ x $\mid$ y $\mid \ldots$ |

Figure 3: A fragment from the context-free grammar of our ARC DSL.

non-terminal, and $\mathcal{R}$ is the set of production rules. An example CFG is shown in Fig. 3. We denote with $\mathcal{R}(N)$ the set of all rules $R \in \mathcal{R}$ whose left-hand side is N. A grammar $\mathcal{G}$ defines a (leftmost) *single-step derivation* relation on sequences of symbols: $sN\alpha \Rightarrow s\beta\alpha$ if $N \rightarrow \beta \in \mathcal{R}$, where $s \in \Sigma^*$ and $\alpha, \beta \in (\mathcal{N} \cup \Sigma)^*$. The transitive closure of this relation $\Rightarrow^*$ is called (leftmost) *derivation*.

**Programs** A *program* $P \in \Sigma^*$ is a terminal sequence derivable from some $N \in \mathcal{N}$; we call a program *whole* if it is derivable from $\mathcal{S}$. The set of all programs is called the *language* of the grammar $\mathcal{G}$: $\mathcal{L}(\mathcal{G}) = \{s \in \Sigma^* \mid N \Rightarrow^* s\}$. The *trace* of a program $\text{tr}(P)$ is the sequence of production rules $R_1, \ldots, R_n$ used in its derivation ($N \Rightarrow \alpha_1 \Rightarrow \ldots \Rightarrow \alpha_{n-1} \Rightarrow P$). The *size* of a program $|P|$ is the length of its trace. The semantics of a program $P$ is defined by the evaluation function $[\![P]\!] : \text{Val}^* \rightarrow \text{Val}$, which maps the values of program variables to its output value.

**Problem Statement** A PBE problem is defined by a DSL with a grammar $\mathcal{G}$ and an evaluation function $[\![\cdot]\!]$, as well as a set of input-output examples $\mathcal{E} = \overrightarrow{\langle i, o \rangle}$ where $i \in \text{Val}^*$, $o \in \text{Val}$. A *solution* to the problem is a program $P \in \mathcal{L}(\mathcal{G})$ such that $\forall \langle i, o \rangle \in \mathcal{E}, [\![P]\!](i) = o$.

## 2.2 Assigning Costs to Programs

**Weighted Context-free Grammar** A *weighted context-free grammar* (WCFG) $\mathcal{G}_w$ is a pair of a CFG $\mathcal{G}$ and a function $w_{\mathbb{R}} : \mathcal{R} \rightarrow \mathbb{R}^+$ that maps each production rule $R \in \mathcal{R}$ to a positive weight. Given a weighted grammar $\mathcal{G}_w$, we can define the *real cost* of a program $P$ as the sum of weights of all the productions in its trace: $\text{cost}_{\mathbb{R}}(P) = \sum_{R_i \in \text{tr}(P)} w_{\mathbb{R}}(R_i)$.

For the purposes of search, it is convenient to define a *discrete weight* function $w : \mathcal{R} \rightarrow \mathbb{Z}^+$, which rounds weights up to the nearest integer: $w(R) = \lceil w_{\mathbb{R}}(R) \rceil$. The (discrete) *cost* of a program $P$ is defined as the sum of discrete production weights: $\text{cost}(P) = \sum_{R_i \in \text{tr}(P)} w(R_i)$. Note that because of error accumulation, the discrete cost of a program can differ from its rounded real cost, but the difference can be made arbitrarily small by scaling all the costs by a constant factor $\alpha > 1$.

**Probabilistic Context-free Grammar** A popular way to assign weights to production rules is via a *probabilistic context-free grammar* (PCFG). A PCFG $\mathcal{G}_p$ is a pair of a CFG $\mathcal{G}$ and a function $p : \mathcal{R} \rightarrow [0, 1]$ that maps each production rule $R \in \mathcal{R}$ to its probability, such that probabilities of all the rules for a given non-terminal $N \in \mathcal{N}$ sum up to one: $\forall N. \sum_{R \in \mathcal{R}(N)} p(R) = 1$. A PCFG defines a probability distribution on programs: $p(P) = \prod_{R_i \in \text{tr}(P)} p(R_i)$.

Given a PCFG $(\mathcal{G}, p)$ we can derive a WCFG $\mathcal{G}_w$ where $w_{\mathbb{R}}(R) = -\log(p(R))$; to make sure that all weights are finite and positive, we exclude rules with $p(R) = 0$ and inline rules with $p(R) = 1$. In this WCFG, the real cost of a program is related to its probability: $\text{cost}_{\mathbb{R}}(P) = -\log(p(P))$.

## 2.3 Bottom-up Search

Bottom-up search is a popular search technique in program synthesis [2, 11, 28, 36, 39], which enumerates programs from the DSL in the order of increasing costs until it finds a program that satisfies the given examples. The search is implemented as a dynamic programming algorithm (see Alg. 1), which maintains a program *bank* B mapping discrete costs to programs of that cost. Starting with an empty bank and current cost level $\text{LVL} = 1$, the search iteratively creates all programs of cost 1, 2, 3, and so on; to create complex programs, the algorithm *reuses* simpler programs already stored in the bank, and combines them using the production rules of the grammar.

For example, consider the CFG in Fig. 3, and assume a uniform weight function $w(\cdot) = 1$. Then in the first iteration (cost level 1), the algorithm will enumerate programs consisting of a single literal or

**Algorithm 1** Bottom-Up Search Algorithm

**Input:** Input-output examples $\mathcal{E}$, a WCFG $\mathcal{G}_w = (\mathcal{N}, \Sigma, \mathcal{S}, \mathcal{R}, w)$
**Output:** A program $P$ consistent with $\mathcal{E}$ or failure ($\bot$)
1: **procedure** BOTTOM-UP-SEARCH($\mathcal{G}_w, \mathcal{E}$)
2:     LVL, B, E $\leftarrow 1, \emptyset, \emptyset$                                                 ▷ Initialize state of the search
3:     **while** true **do**
4:         **for** $P \in$ NEW-PROGRAMS($\mathcal{G}_w$, LVL, B) **do**         ▷ For all programs of cost LVL
5:             EVAL $\leftarrow [\langle i, [\![P]\!](i) \rangle \mid \langle i, o \rangle \in \mathcal{E}]$           ▷ Evaluate on inputs from $\mathcal{E}$
6:             **if** (EVAL $= \mathcal{E}$) **then**
7:                 **return** $P$                         ▷ $P$ fully satisfies $\mathcal{E}$, solution found!
8:             **else if** (EVAL $\in$ E) **then**
9:                 **continue**             ▷ $P$ is semantically equivalent to another program in B
10:             B[LVL] $\leftarrow$ B[LVL] $\cup \{P\}$           ▷ Add to the bank, indexed by cost
11:             E $\leftarrow$ E $\cup$ EVAL                      ▷ Cache evaluation result
12:     LVL $\leftarrow$ LVL $+ 1$
13:     **return** $\bot$                                      ▷ Cost limit reached
14: **procedure** NEW-PROGRAMS($\mathcal{G}_w$, LVL, B)
15:     **for** R $= N \rightarrow s_0 N_1 s_1 N_2 \ldots N_k s_k \in \mathcal{R}$ **do**     ▷ R is a production rule with $k$ non-terminals
16:         **for** $(c_1, \ldots, c_k) \in \left\{ [1..\text{LVL} - 1]^k \mid \sum c_i = \text{LVL} - w(R) \right\}$ **do**    ▷ For all subexpression costs
17:             **for** $(P_1, \ldots, P_k) \in \left\{ \text{B}[c_1] \times \ldots \times \text{B}[c_k] \mid \bigwedge_i N_i \Rightarrow^* P_i \right\}$ **do**    ▷ For all subexpressions
18:                 **yield** $s_0 P_1 s_1 P_2 \ldots P_k s_k$          ▷ Substitute subexpressions into R's RHS

variable—*e.g.* `self`, GREY, UP, *etc*—and store them in B[1]. At cost level 2, it will enumerate unary operators applied to programs stored in B[1]: *e.g.* `color_of(self)`, `move(UP)`, *etc*. More generally, at cost level LVL, the algorithm considers all available productions, and for each production, enumerates all combinations of arguments whose costs sum up to LVL $- 1$.

During search, each candidate expression is evaluated to see if it satisfies the examples (lines 5–7). Importantly, the search maintains a cache of all evaluation results E, and discards the newly constructed program if it is *observationally equivalent* to a program already in the bank (line 8), *i.e.* if it evaluates to the same output for all inputs in the examples. This step is the key to the efficiency of the bottom-up search algorithm: it allows the synthesizer to factorize the search space by evaluation result, significantly reducing the number of programs explored at each cost level.

## 3 The HYSYNTH Approach

A key challenge in program synthesis is the astronomical size of the search space the synthesizer has to explore. For example, to find the program Eq. 1, the solution to the ARC task from the introduction, bottom-up search with a uniform weight function has to enumerate around 450K programs (all programs of size $\leq 16$), which takes 4.5 minutes in our experiments.

On the other hand, sampling solutions to this task from an LLM yields programs that are *close* to the desired solution, even if not quite correct. As we show in Appendix A, GPT-4o uses relevant components `update_color`, `color_of`, and `is_neighbor` in nearly all of its solutions (usually missing some part of the filter or using the wrong color in the transform), and never uses irrelevant components like `move` or `rotate`. This suggests that the LLM generally has the right intuition about the components the solution needs to use; our insight is to leverage this intuition to guide bottom-up search by *assigning lower weights to the components that the LLM uses frequently*.

### 3.1 Guiding Bottom-up Search with Context-Free LLM Approximation

The overview of our approach, HYSYNTH, is shown in Fig. 2. Given a PBE problem consisting of a DSL with grammar $\mathcal{G}$ and a set of input-output examples $\mathcal{E}$, HYSYNTH proceeds in three steps.

**Step 1: Sampling Solutions from an LLM** HYSYNTH starts by creating an LLM prompt that contains $\mathcal{G}$ and $\mathcal{E}$; the prompt can be optionally augmented with in-context examples if they are available for the given DSL. A complete prompt for the ARC running example can be found in Appendix B. The LLM is then used to sample a set $\{S_i\}_{i=1}^N$ of completions; the choice of $N$ trades off computational cost and the faithfulness of the approximation to the true LLM conditional.

**Step 2: Learning a PCFG from LLM Solutions** Next, HYSYNTH attempts to parse each completion $S_i$ into a program $P_i$ using the grammar $\mathcal{G}$. The resulting set of programs $\{P_i\}_{i=1}^{N'}$ (where $N' \leq N$) is used to learn a PCFG $\mathcal{G}_p$ via maximum likelihood estimation: $p(\text{R}) = \frac{\text{count}(\text{R})+\alpha}{\sum_{\text{R} \in \mathcal{R}} \text{count}(\text{R})+\alpha \times |\mathcal{R}|}$. Here $\text{count}(\text{R})$ is the frequency of rule R in all the derivations of the programs in $\{P_i\}$ and $\alpha$ is a smoothing parameter that ensures that every rule has a non-zero probability (typically set to 1).

Our experiments show that some models struggle to generate grammatical completions, leading to $N' \ll N$. To increase the sampling efficiency in those cases, HYSYNTH implements *non-strict mode*, where ungrammatical completions $S_i$ are not discarded. Instead the tool performs lexical analysis on $S_i$ to convert it into a sequence of terminals and approximates the frequency of each production R based on the frequency of its *operator terminal*, a designated terminal of R, which represents a DSL operator; *e.g.* $\text{count}(Atom \rightarrow \texttt{not}\ Atom) = \text{count}(\texttt{not})$.[3]

**Step 3: Guiding Bottom-up Search with PCFG** Finally, HYSYNTH uses the PCFG computed in the previous step to derive a weighted grammar $\mathcal{G}_w$ as explained in Sec. 2.2, and uses it to initialize the bottom-up search procedure in Alg. 1. As a result, the search is guided by the insights from the the LLM. For example, the WCFG learned from the GPT-4o completions for the ARC task above gives the relevant transform operator `update_color` weight 2, while all other *Transform* rules have weight 4; the relevant filter operators `color_of` and `is_neighbor` are similarly down-weighted. As a result, the search procedure only has to enumerate around 220K programs instead of 450K, achieving a 4x speedup, and solving the motivating example in just one minute with LLM guidance.

## 3.2 Domain-Specific Instantiations

We now describe how the HYSYNTH approach is instantiated in three different domains: ARC grid puzzles, TENSOR manipulations, and STRING manipulations.

**ARC Domain** An example task from this domain is shown in Fig. 1a and has been used as a running example throughout this paper. There is no established DSL for ARC, and arguably, DSL design is the biggest challenge when attempting to solve ARC using a PBE approach, since it is hard to capture the wide variety of tasks in this domain. Our DSL is inspired by the rule-based language of ARGA [44], which we modified slightly to make it more compositional.

A program in our DSL is a sequence of rules of the form **if** *filter* **then** *transform*. A rule refers to the current object **self**, which is modified by the transform if the filter is satisfied in the current state of the grid. The rule can also refer to other objects in the grid, such as `other` in Eq. 1. This program is well-defined because its filter uniquely identifies the object `other`; if the filter is too weak to uniquely determine the effect of the transform, the program's output is considered undefined. The full grammar of our DSL can be found in Appendix H.

Instead of searching for a complete program using Alg. 1, we further optimize our synthesizer using a divide-and-conquer strategy inspired by [6], searching for filters and transforms *separately*. Specifically, HYSYNTH-ARC first searches for transforms that are correct on some objects in the grid; once it has found a set of transforms that collectively describe all grid objects, it searches for filters that distinguish between the subsets of objects changed by each transform.

Consider once again our running example. When the transform synthesizer enumerates the expression `update_color(color_of(other))`, it detects that this transform works for all *grey objects*, because for each grey object **self** there exists a corresponding object `other` whose color can be copied. Now the goal of filter synthesis is to find a boolean expression that holds exactly for those pairs of objects (**self**, `other`) that make the transform work. See Appendix K for more details about this algorithm.

**TENSOR Domain** This domain originates from the TFCODER synthesizer [36], which takes as input examples of a tensor transformation (with an optional natural language description) and synthesizes a TensorFlow program that performs the transformation. An example task from this domain is shown in Fig. 1b, whose solution is: `tf.gather_nd(in1, tf.stack((in2, in3), axis=-1))`. The main challenge, however, is that the TensorFlow grammar is very large (see Appendix G), and most importantly, the programs are allowed to use an *unbounded* set of constants. The original TFCODER

---

[3]Typically, the operator terminal uniquely identifies R, but when this is not the case, we can normalize $\text{count}(\text{R})$ by the number of rules in $\mathcal{R}$ that produce this terminal.

synthesizer requires the user to provide any non-standard constants that a task might require, and, according to their paper, this is the main barrier to the usability of their tool.

For program synthesis in this domain we use the TFCODER synthesizer off the shelf. TFCODER performs weighted bottom-up search, using a combination of hand-tuned weights and weights derived by two custom-trained neural models. HYSYNTH-TENSOR replaces these weights entirely with weights computed by sampling from an LLM. Importantly, our version of the tool does not require the user to provide any constants; instead we extract constants from the LLM completions, whereby significantly reducing the burden on the user.

**STRING Domain**    Our third domain involves string manipulation tasks from the SYGUS competition [4], which are inspired by spreadsheet use cases. An example task, which requires extracting the top-level domain name from a URL, is shown in Fig. 1c. In this domain we use the PROBE [11] synthesizer off the shelf. PROBE performs weighted bottom-up search, starting with a uniform grammar and updating the weights on the fly; HYSYNTH-STRING instead initializes PROBE's search with weights derived from an LLM, and disables the weight updates during search.

## 4    Experiments and Results

### 4.1    Experimental Setup

We evaluate HYSYNTH on 299 PBE tasks from three different domains: ARC (160 tasks), STRING (70 tasks) and TENSOR (69 tasks).

**ARC Benchmark**    The 160 ARC tasks are taken from the testing set of ARGA [44]. This *object-centric* subset of the full ARC corpus is known as OBJECT-ARC, and has been used to evaluate other ARC solvers [27]. ARC specifications consist of 2-7 input-output training grids and 1 testing grid. Correctness is based on whether the generated solution produces the correct output on the testing grid. Our ARC DSL has a total of 20 operations and 50 constants and variables across all types.

**TENSOR Benchmark**    The 69 TENSOR tasks taken from TFCODER focus on tensor manipulation. 49 of them are sourced from StackOverflow inquiries, and 20 are from real-world scenarios faced by TensorFlow users at Google. The overall benchmark suite consists of 72 tasks. We use three of these tasks as in-context examples and evaluate on the rest. The grammar for this domain consists of 134 Tensorflow operations, primitives like `0`, `1`, `-1`, `True` and other task-specific constants.

**STRING Benchmark**    The 70 STRING tasks are taken from testing set of PROBE, which is derived from the SYGUS benchmark [4]. The number of examples ranges from 2 to 400. The original SYGUS benchmark have custom grammars for each task, but we use a union of all the grammars to make the search more challenging; the union grammar has 16 operations and 59 constants.

**Configurations**    Our main HYSYNTH configuration uses GPT4O as the LLM, with 100 samples per task to learn a PCFG in non-strict mode (*i.e.* syntactically invalid completions are included in the PCFG learning process, as explained in Sec. 3.1). For each domain, we compare the performance of HYSYNTH with a baseline synthesizer for that domain (ARGA[4], PROBE, and TFCODER), as well as three ablations: (1) *no search*, *i.e.* using the 100 samples from the LLM directly, (2) *unguided search*, *i.e.* running the same synthesizer but with a uniform weighted grammar, and (3) *binary surrogate*, running the synthesizer but with a *binary PCFG*, *i.e.* a CFG that includes the components present in the LLM samples with equal probabilities, and excludes all other components completely. We also analyze the performance of HYSYNTH with different number of samples used to learn the PCFG (10, 20, and 50), with other LLMs (GPT3.5 and DEEPSEEK [22]), as well as in strict mode (which discards syntactically invalid LLM completions). The timeout is set to 10 minutes for all experiments and includes the search time and time to sample LLM completions (and compute PCFG). The average time to sample 100 solutions from GPT4O is 4 seconds, 12 seconds and 20 seconds per task for the STRING, ARC and TENSOR domains, respectively.

### 4.2    Results

**How does HYSYNTH compare to baselines and ablations?**    We compare the time to solution for the main HYSYNTH configuration, baseline synthesizers, and the three ablations; the results

---

[4]At the time of writing, ARGA is no longer state of the art on the OBJECT-ARC dataset; we explain in Sec. 5 why the comparison with ARGA is still relevant.

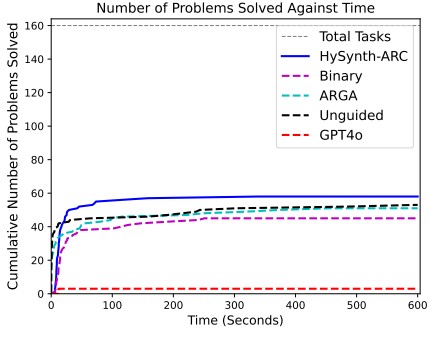

(a) HYSYNTH-ARC results with GPT4O

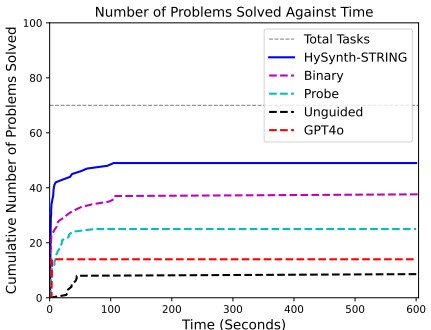

(b) HYSYNTH-STRING results with GPT4O

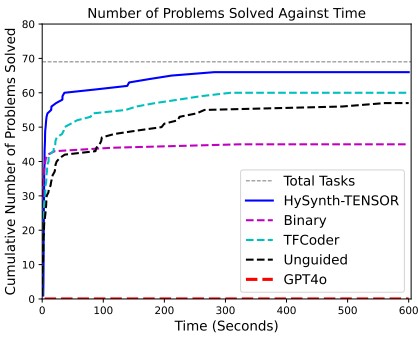

(c) HYSYNTH-TENSOR results with GPT4O

| Domain/Model | % Valid completions |
|---|---|
| TENSOR-GPT4O | 99.9% |
| TENSOR-DEEPSEEK | 92.8% |
| STRING-GPT4O | 37.5% |
| STRING-DEEPSEEK | 0% |
| ARC-GPT4O | 78.4% |

(d) Percentage of syntactically valid completions

Figure 4: (a,b,c) Number of benchmarks solved by HYSYNTH as a function of time for the ARC, TENSOR, and STRING domains; timeout is 10 min. (d) Percentage of syntactically valid completions per domain.

for the three domains are shown in Fig. 4a, Fig. 4b, and Fig. 4c. Overall, HYSYNTH consistently outperforms both the baseline synthesizers and ablations, solving more tasks across all domains.

In more detail, direct LLM sampling performs very poorly on all domains, solving between 0 and 14 tasks; this confirms our hypothesis that LLMs struggle on PBE tasks in domain-specific languages, which are not prevalent in their training data. Interestingly, despite not being able to solve *any* TENSOR tasks by itself, GPT4O provides excellent guidance for HYSYNTH on that domain, helping it solve 96% of the total benchmark! On the other hand, synthesis guided by a binary surrogate model performs worse than HYSYNTH (and even unguided search in case of ARC and TENSOR) since the search excludes essential components from the grammar.

In STRING and TENSOR domains, the baseline synthesizers predictably do better than unguided search, since both use the same search implementation, but with different weights. On ARC, however, our custom synthesizer outperforms ARGA[5] even without LLM guidance; this speaks to the efficiency of the bottom-up search and the divide-and-conquer strategy we use, which are results of years of research in the program synthesis community.

**How many samples are needed to learn a PCFG?** To better understand how the number of samples affects the quality of PCFG guidance, we vary the number of GPT4O programs used in PCFG learning $N = 10, 20, 50, 100$, and once again measure the number of tasks solved over time. The results are shown in Fig. 5a, Fig. 5b, and Fig. 5c. As expected, larger sample sizes generally lead to better performance, but the difference is minimal: in ARC and TENSOR, the difference between the best and worst performing versions of HYSYNTH is only 2 problems each, while in STRING, HYSYNTH solves 9 fewer problems with 10 samples than with 100. Despite these differences, all versions of HYSYNTH still outperform the baseline and unguided search. This suggests that fewer samples are sufficient to effectively train a robust surrogate model, thereby optimizing costs.

<hr />

[5][44] report 57 tasks for ARGA but we could only reproduce 51 on our hardware with a 10 minute timeout.

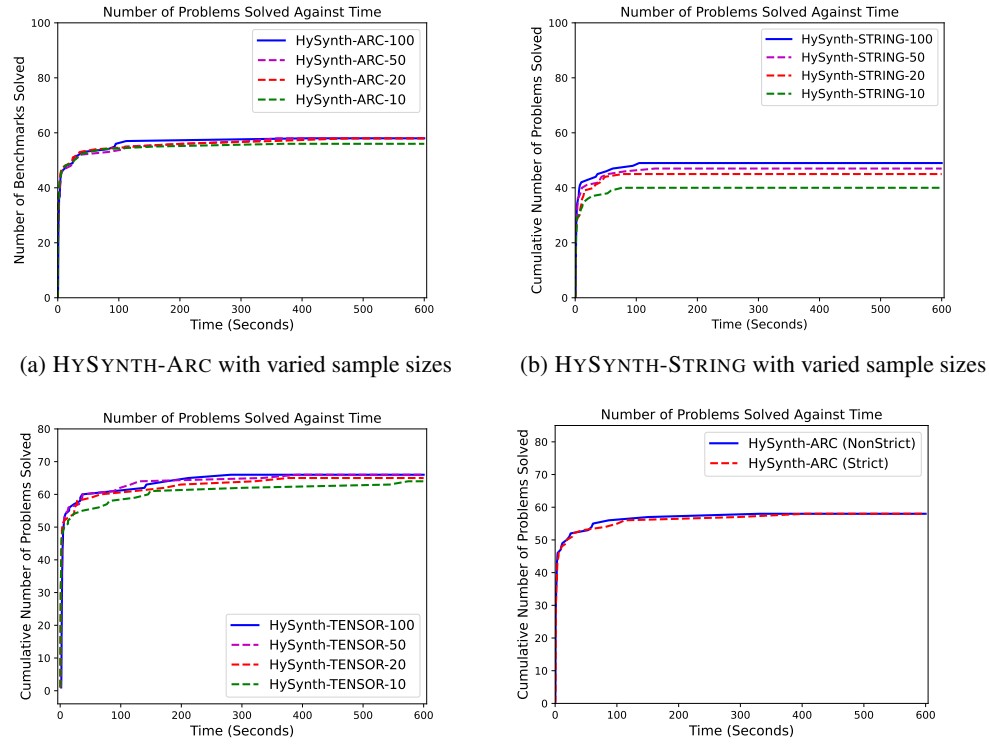

(a) HYSYNTH-ARC with varied sample sizes    (b) HYSYNTH-STRING with varied sample sizes

(c) HYSYNTH-TENSOR with varied sample sizes    (d) HYSYNTH-ARC with strict and non-strict modes

Figure 5: HYSYNTH-ARC, HYSYNTH-TENSOR and HYSYNTH-STRING results guided by a PCFG learned from different number of GPT4O samples (n=10, 20, 50, 100).

**Do our results generalize to other models?** To answer this question, we repeat our experiments on STRING and TENSOR domains with GPT3.5 and the open-source model `deepseek-coder-33b-instruct` (DEEPSEEK) [22]. The results with these models are detailed in Fig. 9 in Appendix C, and they corroborate the pattern observed with GPT4O, where the guided versions outperform the baseline, unguided search, and direct sampling from the LLM.

**How important is non-strict mode?** Fig. 4d shows the percentage of syntactically valid completions generated by GPT4O and DEEPSEEK (where applicable). You can see that while on TENSOR almost all completions are valid, this percentage falls to 78.4% for ARC and 37.5% for STRING; this is not surprising, given that the former are TensorFlow programs, which the model has seen during training, while the latter two are custom DSLs. In the STRING benchmark, the grammar is very restricted (*e.g.* only numeric constants allowed are 0-9), and the LLM has trouble adhering to this restricted grammar. But even if we were to relax the definition of syntactic validity, LLM solutions would achieve a syntactic validity of only 47%. Hence our non-strict mode proves especially helpful for low-resource domains, where otherwise we would have to discard a large proportion of completions. At the same time, we find that *given the same number of completions to learn from*, the PCFGs learned in non-strict mode are just as effective as those learned in strict mode: as shown in Fig. 5d, HYSYNTH-ARC with the guidance from 100 GPT4O completions solves 58 tasks *in either mode* (with the difference that strict mode has to sample more completions to get 100 valid ones).

### 4.3 Limitations

The main limitation of our hybrid approach *wrt.* purely neural approaches is that it requires implementing a synthesizer for each DSL of interest; although we have shown that the same bottom-up search can be used across different domains, some implementation effort is still required. On the other hand, compared to purely symbolic approaches, our method requires sampling from an LLM, which is costly; additionally, the guidance provided by our approach is only as good as the LLM's completions: if they contain many irrelevant operators, our guided search can be *slower* than unguided

search. Finally, our experiments are subject to the usual threat that the LLMs might have seen our benchmarks in their training data; we do not consider it a major issue, however, given that our main result is the superior performance of guided search *relative* to using LLMs without search.

## 5 Related Work

**Guiding Program Synthesis with Probabilistic Models**   The traditional approach to *program synthesis* is based on combinatorial search [7], augmented with pruning techniques based on program semantics [2, 6, 39]. To further speed up the search, researchers have proposed *guiding* the search with a learned probabilistic model. Most approaches to guided search use special-purpose models that have to be trained on a domain-specific corpus of programs [26] or PBE tasks [9, 24, 32, 37]. Although some of these models can be trained on synthetic data, the training process is still expensive and requires manual tuning, which makes it hard to apply these techniques to new domains.

With the advent of pretrained Large Language Models (LLMs), it seems only natural to use them to guide search-based program synthesis, thus alleviating the need for domain-specific training data. We are only aware of one other attempt to do this: concurrent work by Li et al. [29], which also extracts a PCFG from the LLM's samples, similarly to HySynth. An important difference is that they use the PCFG to guide *top-down* A* search, while we use it to guide *bottom-up* search, which is known to be more efficient (they also evaluate their tool on synthesis from logical formulas as opposed to PBE).

**Solving the Abstraction and Reasoning Corpus**   All state-of-the-art solvers for this benchmark have relied on carefully curated DSLs for Arc [3, 13, 19, 27, 43]. Xu et al. [44] proposed the DSL we extend in our approach, and the Object-Arc subset we evaluate on. Lei et al. [27] embed their DSL as a subset of PDDL and use a Generalized Planning (GP) algorithm as their search component. They have the current best performance on Object-Arc, however they encode more domain-knowledge in the form of preconditions and per-abstraction restrictions on filters and transforms, to make GP viable. Our approach does not require this additional information. [3, 10] use DreamCoder [16], to perform execution-guided search over a DSL for grid manipulations, however they only provide proof-of-concept evaluations. [38, 41] also use an LLM to generate code given the spec of the task. Both of these approaches interact with the model across several rounds, while our technique uses the suggestions from the LLM only as a starting point. Our technique also performs a complete search guided by the LLM distribution, enabled by the structure of our DSL, whereas previous approaches only consider code directly generated by the LLM.

## 6 Conclusion and Future Work

Our approach introduces a robust technique for using both valid and invalid completions from an LLM to learn a surrogate model. By incorporating ungrammatical completions, we can extract useful insights that would otherwise be discarded. Overall, we provide an alternative to the conventional strategy of large-scale sampling from LLMs, proposing a more effective use of the available completions to guide the search process. An interesting future direction would be to guide search with a more expressive context-dependent surrogate model.

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

```
// Solution 1, occurs 6 times
if color_of(self) = GREY ∧ is_neighbor(self, other)
        then update_color(color_of(other))

// Solution 2, occurs 1 time
if is_neighbor(self, other) ∧ color_of(other) = GREY
        then update_color(color_of(other))

// Solution 3, occurs 1 time
if color_of(self) = GREY
        then update_color(color_of(other))

// Solution 4, occurs 1 time
if not (color_of(self) = GREY) ∧ is_neighbor(self, other) ∧ color_of(other) = GREY
        then update_color(FUCHSIA)

// Solution 5, occurs 1 time
if size_of(self) = 4 then update_color(RED) ;
if size_of(self) = 4 ∧ color_of(self) = GREY then update_color(FUCHSIA) ;
if size_of(self) = 4 ∧ color_of(self) = BLUE then update_color(ORANGE) ;
if size_of(self) = 4 ∧ color_of(self) = YELLOW then update_color(CYAN)
```

Figure 6: Ten samples from GPT4o for the motivating example in Fig. 1a

## A    GPT4o Solutions for the Motivating Example

Recall the motivating example in Fig. 1a where the task is to update the color of the grey objects to the color of their single-pixel neighbor. As a reminder, the smallest correct solution to this task consists of the following rule:

```
if color_of(self) = GREY ∧ is_neighbor(self, x) ∧ size_of(x) = MIN
        then update_color(color_of(x))
```

Fig. 6 shows the programs we obtained by deduplicating 10 samples from GPT4o for this task. The syntax of the solutions is slightly modified for readability; our implementation uses a LISP-style s-expression syntax [30] to simplify parsing.

As you can see, the most frequent solution is almost correct, except that it does not constrain the neighbor `other` to be of size 1; this leads to the constraint being ambiguous (since every grey object has multiple neighbors of different colors), in which case the program semantics is considered undefined. That said, you can observe that the model consistently uses relevant components, such as `color_of`, `is_neighbor`, and `update_color`, which enables us to extract a useful PCFG from these solutions.

When we increased the sample size to 125, GPT4o was able to produce one correct solution (which is slightly larger than the minimal solution above):

```
if color_of(self) = GREY ∧ is_neighbor(self, other) ∧ not (color_of(other) = GREY)
        then update_color(color_of(other))
```

```
You are an assistant chatbot with human-like perception, reasoning and learning capabilities.
You can solve tasks concisely, efficiently, and moreover, correctly.
Let's engage in perception and logic-based tasks.
You only output source code.
No explanations or any other text.
```

Figure 7: System prompt for ARC domain.

## B    LLM Prompt for the ARC Grammar

### B.1    System Prompt

The system prompt given to the LLM for ARC domain is shown in Fig. 7.

### B.2    User Prompt

The full user prompt for the ARC domain is shown in Fig. 8. It contains the domain-specific language, four in-context examples and the query for the test task.

You are an efficient assistant for logical reasoning and code generation.
You will help me solve a visual perception and reasoning task.
I will first provide you with the definition of a Domain Specific Language you will use
   ↪ for writing a solution for the task.
I will then present you with the description of the task that you will be tested in.
You will then respond to the queries I make regarding the solution of the task.

This is the definition of the DSL you will use to solve the task.
It is given as a context-free grammar in the EBNF format used by the Lark parser
   ↪ generator, with some informative comments about the semantics.
You will return a string that is parseable by the 'program' non-terminal of the grammar.

```
library: "(" program* ")"

// Rules are executed one after another, in the order they appear.
// There could be no rules, in which case the program does nothing.
program: "(" "do" rule* ")"
...
```

⋘ DSL IMPLEMENTATION IN LARK ⋙

```
Now we continue with the visual perception and reasoning task.
The input for the task is a small number of pairs of grids of characters.
The value of each of the cells of the grids are the colors defined in the DSL, so we can
   ↪  think of grids as images.
Each pair of images correspond to an input-output example for an unknown program P.
For each pair, the program P is evaluated on the image grid and operates on the objects
   ↪ that appear in it.
The output of the program is then the output image.
The objects in the images are easy and natural to identify for humans, so there is no
   ↪ need to define them explicitly.
However you are able to abstract them correctly, and the DSL is interpreted with the
   ↪ same correct abstraction.

Now I will show you some demonstration tasks along with the output you would be expected
   ↪  to produce for each of them.

## DEMONSTRATION TASK 1

### INPUT
PAIR 1
INPUT GRID:
0 0 0 0 0 0 0 0
0 0 0 0 0 R 0 0
0 R 0 0 0 R 0 R
0 R R 0 0 R 0 0
0 0 0 0 0 0 0 0
0 R R 0 0 0 0 0
0 R R 0 R R 0 0
0 0 0 0 0 0 0 0
```

Figure 8: User prompt for ARC domain.

OUTPUT GRID:
O O O O O O O O
O O O O O Y O O
O Y O O O Y O Y
O Y Y O O Y O O
O O O O O O O O
O Y Y O O O O O
O Y Y O Y Y O O
O O O O O O O O

≪≪ ENCODING OF EXAMPLE PAIR 2 AND 3 OF DEMO TASK 1≫≫

### EXPECTED OUTPUT
{
    "nl_description": "Recolor all objects to color Y",
    "code": ≪≪ EXPECTED CODE IN DSL ≫≫
}

≪≪ MORE DEMONSTRATION TASKS (4 IN TOTAL) ≫≫

Now follows task you will be evaluated on.
Output the solution as a JSON object, which should contain both a natural language
    ↪ description of the solution and the solution written in the DSL.
The code should be parseable by the DSL grammar.
The JSON must have the following structure:

{
    "nl_description": "TO_BE_FILLED",
    "code": "TO_BE_FILLED"
}

## TEST TASK

PAIR 1
INPUT GRID:
O O R O O F O O O C
O O O O O O O O O O
O O O O X X X X O O
O O O O X X X X O O
O X X O X X X X O O
O X X O X X X X O O
O X X O O O O O O O
O X X O O O O X X X
O X X O O O O X X X
O O O O O O O O X X X
OUTPUT GRID:
O O R O O F O O O C
O O O O O O O O O O
O O O O F F F F O O
O O O O F F F F O O
O R R O F F F F O O
O R R O F F F F O O
O R R O O O O O O O
O R R O O O O C C C
O R R O O O O C C C
O O O O O O O O C C C

≪≪ REST OF THE I/O EXAMPLES OF TEST TASK ≫≫

# C  Experimental results with LLMs DEEPSEEK and GPT3.5

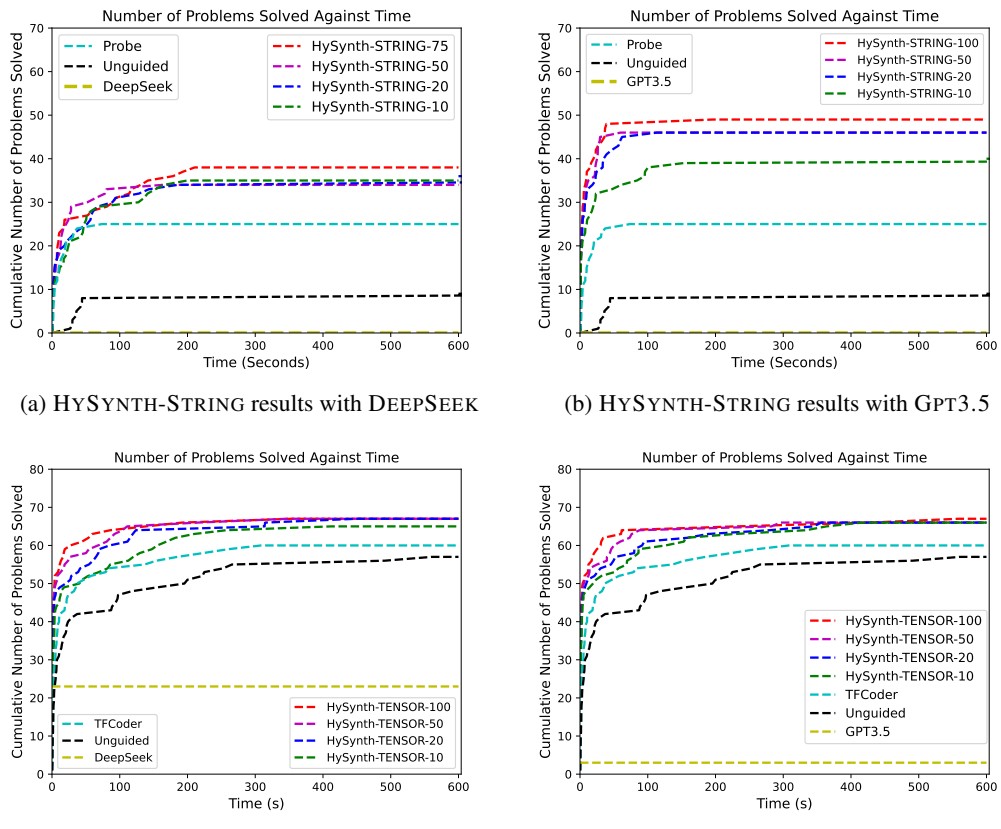

(a) HYSYNTH-STRING results with DEEPSEEK

(b) HYSYNTH-STRING results with GPT3.5

(c) HYSYNTH-TENSOR results with DEEPSEEK

(d) HYSYNTH-TENSOR results with GPT3.5

Figure 9: HYSYNTH-STRING and HYSYNTH-TENSOR evaluation results with DEEPSEEK and GPT3.5.

```
You are a coding assistant. Be precise and terse.
You will be provided a list of tensorflow operators, a task description, and some input/output examples.
Your task is to generate the body of a python function that will transform the input to the output.
Only use the operators provided in the list.
Your answer should be as short as possible while still being correct.
Make sure to only generate python code.
```

Figure 10: System prompt for TENSOR domain.

```
[TENSORFLOW OPERATORS]
⋘ see appendix E ⋙

[TASK DESCRIPTION]
index into the tensor

[INPUTS]
[[ 5.  2.]
 [ 1.  3.]
 [ 0. -1.]]

[OUTPUTS]
[[[ 5.  5.]
  [ 1.  1.]
  [ 0.  0.]]

 [[ 2.  2.]
  [ 3.  3.]
  [-1. -1.]]]

[PROGRAM]
def transform(in1):
```

Figure 11: User prompt for TENSOR domain

# D  LLM Prompt for the TENSOR Grammar

The system and user prompt for TENSOR domain are in Fig. 10 and Fig. 11.

```
You are a coding assistant. Be precise and terse.
You will be given a SyGuS grammar, a natural language specification, and a set of input-
    ↪ output examples.
Your task is to complete the provided function definition with an implementation that is
    ↪  correct according to the grammar, specification, and examples.
Your answer should be as short as possible while still being correct.
Make sure that your answer is a valid s-expression.
```

Figure 12: System prompt for STRING domain

```
[GRAMMAR]
(synth-fun f ((_arg_0 String)) String ((Start String (ntString)) (ntString String (
    ↪ _arg_0 "" " " "BRD" "DRS" "LDS" "Branding" "Direct Response" "Leads" "=" "/" "in
    ↪ " "_" "9" "." "microsoft" "windows" "apple" "mac" "-" "1" "2" "3" "4" "5" "6"
    ↪ "7" "8" "0" "," "<" ">" "/n" "%" "b" "apple" "bananas" "strawberries" "oranges"
    ↪ "LLC" "Inc" "Corporation" "Enterprises" "Company" "(" ")" "+" "name" "," (str.++
    ↪  ntString ntString) (str.replace ntString ntString ntString) (str.at ntString
    ↪ ntInt) (int.to.str ntInt) (ite ntBool ntString ntString) (str.substr ntString
    ↪ ntInt ntInt))) (ntInt Int (-1 1 2 3 4 5 6 7 8 9 0 1 0 -1 (+ ntInt ntInt) (-
    ↪ ntInt ntInt) (str.len ntString) (str.to.int ntString) (ite ntBool ntInt ntInt) (
    ↪ str.indexof ntString ntString ntInt))) (ntBool Bool (true false (= ntInt ntInt)
    ↪ (str.prefixof ntString ntString) (str.suffixof ntString ntString) (str.contains
    ↪ ntString ntString)))))

[NATURAL LANGUAGE SPECIFICATION]
; https=//exceljet.net/formula/get-top-level-domain-tld

[EXAMPLES]
www.domain.com → com
mail.net → net
www.amazon.co.uk → uk

[SOLUTION]
(define-fun f (_arg_0 String) String
```

Figure 13: User message for STRING

# E    LLM Prompt for STRING

The system and user prompt for STRING domain are in Fig. 12 and Fig. 13.

# F    The Full STRING Grammar

The full grammar for the STRING domain is detailed in Fig. 14.

$$
\begin{array}{rll}
Start \rightarrow & S \\
S \rightarrow & \texttt{arg0} \mid \texttt{arg1} \mid \ldots & \text{string variables} \\
\mid & \texttt{lit-1} \mid \texttt{lit-2} \mid \ldots & \text{string literals} \\
\mid & (\texttt{replace}\ S\ S\ S) & \texttt{replace s x y} \text{ replaces first occurrence of } \texttt{x} \text{ in } \texttt{s} \text{ with } \texttt{y} \\
\mid & (\texttt{concat}\ S\ S) & \texttt{concat x y} \text{ concatenates } \texttt{x} \text{ and } \texttt{y} \\
\mid & (\texttt{substr}\ S\ I\ I) & \texttt{substr x y z} \text{ extracts substring of length } \texttt{z}, \text{ from index } \texttt{y} \\
\mid & (\texttt{ite}\ B\ S\ S) & \texttt{ite x y z} \text{ returns } \texttt{y} \text{ if } \texttt{x} \text{ is true, otherwise } \texttt{z} \\
\mid & (\texttt{int.to.str}\ I) & \texttt{int.to.str x} \text{ converts int } \texttt{x} \text{ to a string} \\
\mid & (\texttt{at}\ S\ I) & \texttt{at x y} \text{ returns the character at index } \texttt{y} \text{ in string } \texttt{x} \\
B \rightarrow & \texttt{true} \mid \texttt{false} & \text{bool literals} \\
\mid & (\texttt{=}\ I\ I) & \texttt{= x y} \text{ returns true if } \texttt{x} \text{ equals } \texttt{y} \\
\mid & (\texttt{contains}\ S\ S) & \texttt{contains x y} \text{ returns true if } \texttt{x} \text{ contains } \texttt{y} \\
\mid & (\texttt{suffixof}\ S\ S) & \texttt{suffixof x y} \text{ returns true if } \texttt{x} \text{ is the suffix of } \texttt{y} \\
\mid & (\texttt{prefixof}\ S\ S) & \texttt{prefixof x y} \text{ returns true if } \texttt{x} \text{ is the prefix of } \texttt{y} \\
I \rightarrow & \texttt{arg0} \mid \texttt{arg1} \mid \ldots & \text{int variables} \\
\mid & \texttt{lit-1} \mid \texttt{lit-2} \mid \ldots & \text{int literals} \\
\mid & (\texttt{str.to.int}\ S) & \texttt{str.to.int x} \text{ converts string } \texttt{x} \text{ to a int} \\
\mid & (\texttt{+}\ I\ I) & \texttt{+ x y} \text{ sums } \texttt{x} \text{ and } \texttt{y} \\
\mid & (\texttt{-}\ I\ I) & \texttt{- x y} \text{ subtracts } \texttt{y} \text{ from } \texttt{x} \\
\mid & (\texttt{length}\ S) & \texttt{length x} \text{ returns length of } \texttt{x} \\
\mid & (\texttt{ite}\ B\ I\ I) & \texttt{ite x y z} \text{ returns } \texttt{y} \text{ if } \texttt{x} \text{ is true, otherwise } \texttt{z} \\
\mid & (\texttt{indexof}\ S\ S\ I) & \texttt{indexof x y z} \text{ returns index of } \texttt{y} \text{ in } \texttt{x}, \text{ starting at index } \texttt{z} \\
\end{array}
$$

Figure 14: The full SYGUS STRING grammar of the PROBE benchmark suite. Integer and string variables and constants change per benchmark. Some benchmark files contain a reduced grammar.

# G The Full TENSOR Grammar

```
General TensorFlow functions:
-----------------------------
tf.abs(x)
tf.add(x, y)
tf.add_n(inputs)
tf.argmax(input, axis)
tf.argmin(input, axis)
tf.argsort(values, axis, stable=True)
tf.argsort(values, axis, direction='DESCENDING', stable=True)
tf.boolean_mask(tensor, mask)
tf.broadcast_to(input, shape)
tf.cast(x, dtype)
tf.clip_by_value(t, clip_value_min, clip_value_max)
tf.concat(values, axis)
tf.constant(value)
tf.constant(value, dtype)
tf.divide(x, y)
tf.equal(x, y)
tf.exp(x)
tf.expand_dims(input, axis)
tf.eye(num_rows)
tf.eye(num_rows, num_columns)
tf.eye(num_rows, dtype)
tf.fill(dims, value)
tf.gather(params, indices)
tf.gather(params, indices, axis, batch_dims)
tf.gather_nd(params, indices)
tf.gather_nd(params, indices, batch_dims)
tf.greater(x, y)
tf.greater_equal(x, y)
tf.math.bincount(arr)
tf.math.ceil(x)
tf.math.count_nonzero(input)
tf.math.count_nonzero(input, axis)
tf.math.cumsum(x, axis)
tf.math.cumsum(x, axis, exclusive=True)
tf.math.divide_no_nan(x, y)
tf.math.floor(x)
tf.math.log(x)
tf.math.negative(x)
tf.math.reciprocal(x)
tf.math.reciprocal_no_nan(x)
tf.math.segment_max(data, segment_ids)
tf.math.segment_mean(data, segment_ids)
tf.math.segment_min(data, segment_ids)
tf.math.segment_prod(data, segment_ids)
tf.math.segment_sum(data, segment_ids)
tf.math.squared_difference(x, y)
tf.math.top_k(input, k)
tf.math.unsorted_segment_max(data, segment_ids, num_segments)
tf.math.unsorted_segment_mean(data, segment_ids, num_segments)
tf.math.unsorted_segment_min(data, segment_ids, num_segments)
tf.math.unsorted_segment_prod(data, segment_ids, num_segments)
tf.math.unsorted_segment_sum(data, segment_ids, num_segments)
```

Figure 15: List of TensorFlow operations as used in TFCODER.

```
tf.matmul(a, b)
tf.maximum(x, y)
tf.minimum(x, y)
tf.multiply(x, y)
tf.not_equal(x, y)
tf.one_hot(indices, depth)
tf.ones(shape)
tf.ones_like(input)
tf.pad(tensor, paddings, mode='CONSTANT')
tf.pad(tensor, paddings, mode='CONSTANT', constant_values)
tf.pad(tensor, paddings, mode='REFLECT')
tf.pad(tensor, paddings, mode='SYMMETRIC')
tf.range(start)
tf.range(start, limit, delta)
tf.reduce_any(input_tensor, axis)
tf.reduce_max(input_tensor)
tf.reduce_max(input_tensor, axis)
tf.reduce_mean(input_tensor)
tf.reduce_mean(input_tensor, axis)
tf.reduce_min(input_tensor)
tf.reduce_min(input_tensor, axis)
tf.reduce_prod(input_tensor, axis)
tf.reduce_sum(input_tensor)
tf.reduce_sum(input_tensor, axis)
tf.reshape(tensor, shape)
tf.reverse(tensor, axis)
tf.roll(input, shift, axis)
tf.round(x)
tf.searchsorted(sorted_sequence, values, side='left')
tf.searchsorted(sorted_sequence, values, side='right')
tf.sequence_mask(lengths)
tf.sequence_mask(lengths, maxlen)
tf.shape(input)
tf.sign(x)
tf.sort(values, axis)
tf.sort(values, axis, direction='DESCENDING')
tf.sqrt(x)
tf.square(x)
tf.squeeze(input)
tf.squeeze(input, axis)
tf.stack(values, axis)
tf.subtract(x, y)
tf.tensordot(a, b, axes)
tf.tile(input, multiples)
tf.transpose(a)
tf.transpose(a, perm)
tf.unique_with_counts(x)
tf.unstack(value, axis)
tf.where(condition)
tf.where(condition, x, y)
tf.zeros(shape)
tf.zeros_like(input)

SparseTensor functions:
-----------------------
tf.SparseTensor(indices, values, dense_shape)
tf.sparse.add(a, b)
tf.sparse.concat(axis, sp_inputs)
tf.sparse.expand_dims(sp_input, axis)
```

```
tf.sparse.from_dense(tensor)
tf.sparse.maximum(sp_a, sp_b)
tf.sparse.minimum(sp_a, sp_b)
tf.sparse.reduce_max(sp_input, axis, output_is_sparse)
tf.sparse.reduce_sum(sp_input, axis, output_is_sparse)
tf.sparse.reset_shape(sp_input)
tf.sparse.reshape(sp_input, shape)
tf.sparse.retain(sp_input, to_retain)
tf.sparse.slice(sp_input, start, size)
tf.sparse.split(sp_input, num_split, axis)
tf.sparse.to_dense(sp_input)
tf.sparse.to_dense(sp_input, default_value)
tf.sparse.to_indicator(sp_input, vocab_size)
tf.sparse.transpose(sp_input)
tf.sparse.transpose(sp_input, perm)

Python-syntax operations:
------------------------
IndexingAxis1Operation: arg1[:, arg2]
IndexingOperation: arg1[arg2]
PairCreationOperation: (arg1, arg2)
SingletonTupleCreationOperation: (arg1,)
SlicingAxis0BothOperation: arg1[arg2:arg3]
SlicingAxis0LeftOperation: arg1[arg2:]
SlicingAxis0RightOperation: arg1[:arg2]
SlicingAxis1BothOperation: arg1[:, arg2:arg3]
SlicingAxis1LeftOperation: arg1[:, arg2:]
SlicingAxis1RightOperation: arg1[:, :arg2]
TripleCreationOperation: (arg1, arg2, arg3)
```

$$
\begin{aligned}
Rule \rightarrow\ & \textbf{if}\ Filter\ \textbf{then}\ Transforms \\
Transforms \rightarrow\ & Transform\ |\ Transform\ ;\ Transforms \\
Filter \rightarrow\ & Atom\ |\ \textsf{not}\ Atom\ |\ Atom \wedge Filter\ |\ Atom \vee Filter \\
Atom \rightarrow\ & Color =_c Color\ |\ Size =_s Size\ |\ Degree =_d Degree\ |\ Width =_w Width\ |\ Height =_h Height \\
 |\ & Shape =_S Shape\ |\ Row =_r Row\ |\ Column =_C Column\ |\ \texttt{is\_neighbor}(Obj, Obj) \\
Transform \rightarrow\ & \texttt{update\_color}(Color)\ |\ \texttt{move}(Dir)\ |\ \texttt{move\_max}(Dir)\ |\ \texttt{extend}(Dir, Overlap) \\
 |\ & \texttt{rotate}(Angle)\ |\ \texttt{fill\_rectangle}(Color, Overlap)\ |\ \texttt{hollow\_rectangle}(Color) \\
 |\ & \texttt{mirror}(Axis)\ |\ \texttt{add\_border}(Color)\ |\ \texttt{flip}(Axis)\ |\ \texttt{NoOp} \\
Obj \rightarrow\ & \textbf{self}\ |\ \texttt{x}\ |\ \texttt{y}\ |\ \ldots \\
Color \rightarrow\ & \texttt{color\_of}(Obj)\ |\ \texttt{GREY}\ |\ \texttt{RED}\ |\ \texttt{BLACK}\ |\ \texttt{BLUE}\ |\ \texttt{YELLOW}\ |\ \texttt{ORANGE}\ |\ \texttt{BROWN}\ |\ \texttt{GREEN}\ |\ \texttt{GREY}\ |\ \texttt{FUCHSIA}\ldots \\
Dir \rightarrow\ & \texttt{dir\_of}(Obj)\ |\ \texttt{UP}\ |\ \texttt{DOWN}\ |\ \texttt{LEFT}\ |\ \texttt{RIGHT}\ |\ \texttt{UPLEFT}\ |\ \texttt{DOWNLEFT}\ |\ \texttt{UPRIGHT}\ |\ \texttt{DOWNRIGHT}\ldots \\
Axis \rightarrow\ & \texttt{axis\_of}(Obj)\ |\ \texttt{VERTICAL}\ |\ \texttt{HORIZONTAL}\ |\ \texttt{LEFTDIAGONAL}\ |\ \texttt{RIGHTDIAGONAL}\ldots \\
Overlap \rightarrow\ & \texttt{TRUE}\ |\ \texttt{FALSE} \\
Angle \rightarrow\ & \texttt{90}\ |\ \texttt{180}\ |\ \texttt{270} \\
Size \rightarrow\ & \texttt{size\_of}(Obj)\ |\ \texttt{MIN}\ |\ \texttt{MAX}\ |\ \ldots \\
Degree \rightarrow\ & \texttt{degree\_of}(Obj)\ |\ \texttt{MIN}\ |\ \texttt{MAX}\ |\ \ldots \\
Width \rightarrow\ & \texttt{width\_of}(Obj)\ |\ \texttt{MIN}\ |\ \texttt{MAX}\ |\ \ldots \\
Height \rightarrow\ & \texttt{height\_of}(Obj)\ |\ \texttt{MIN}\ |\ \texttt{MAX}\ |\ \ldots \\
Column \rightarrow\ & \texttt{column\_of}(Obj)\ |\ \texttt{MIN}\ |\ \texttt{MAX}\ |\ \ldots \\
Row \rightarrow\ & \texttt{row\_of}(Obj)\ |\ \texttt{MIN}\ |\ \texttt{MAX}\ |\ \ldots \\
Shape \rightarrow\ & \texttt{shape\_of}(Obj)\ |\ \texttt{ENCLOSED}\ |\ \texttt{SQUARE}\ |\ \ldots
\end{aligned}
$$

Figure 16: The full grammar for our ARC DSL, object specific parameters like size, degree change per benchmark.

## H  The Full ARC DSL

The full grammar of our ARC DSL is shown in Fig. 16.

## I  Detailed Prompt Settings

For ARC, we sample completions with temperature 1 and 4000 max tokens. For TENSOR, we use temperature 1 and 4000 max tokens. For SYGUS, we use temperature 0.5 and 4000 max tokens. We use the same settings for all three LLMs. When prompting GPT4O, we set `response_type` to JSON.

## J  Broader Research Impacts

Our technique presents a powerful strategy for harnessing both syntactically valid and invalid outputs from an LLM to learn a surrogate model. Incorporating hallucinatory outputs – often erroneous generated by the model, allows us to extract insights that are discarded in standard practices. Our approach mitigates the need for large-scale sampling of completions from LLMs, promoting a more efficient and effective utilization of these models, saving resources. In addition to improving the cost effectiveness of using LLMs, it also opens up new avenues for enhancing model robustness and adaptability across different domains.

---

**Algorithm 2** ARC Synthesis Algorithm

---

**Input:** A set of input-output example grids $\mathcal{E}$, transform grammar $\mathcal{G}_t$ and filter grammar $\mathcal{G}_f$
**Output:** A solution map $\mathcal{M}$ from each transform to the corresponding filter

1: **procedure** HYSYNTH-ARC($\mathcal{E}, \mathcal{G}_p, \mathcal{G}_t$)
2:     LVL, B $\leftarrow 0, \emptyset$          ▷ Initialize search state
3:     $\mathcal{S}_{\text{llm}} \leftarrow$ LLM($\mathcal{E}$)          ▷ Sample solutions from the LLM
4:     $\mathcal{G}_p, \mathcal{G}_t \leftarrow$ INIT($\mathcal{G}_p, \mathcal{S}_{\text{llm}}$), INIT($\mathcal{G}_t, \mathcal{S}_{\text{llm}}$)      ▷ Initialize both PCFGs using LLM solutions
5:     **while** not timeout **do**
6:         $\mathcal{O} \leftarrow$ TRANSFORM-SEARCH($\mathcal{G}_t, \mathcal{E}$)      ▷ Synthesize transforms that cover all objects
7:         $\mathcal{M} \leftarrow$ FILTER-SEARCH($\mathcal{G}_f, \mathcal{E}, \mathcal{O}$)      ▷ Synthesize filters for the above transforms
8:         **if** $\forall (t, f) \in \mathcal{M}, f \neq \bot$ **then**      ▷ Found a filter for each transform
9:             **return** $\mathcal{M}$      ▷ Return the complete solution

---

---

**Algorithm 3** Transform Synthesis Algorithm

---

**Input:** PCFG $\mathcal{G}_t$ and input-output grids $\mathcal{E}$
**Output:** A map $\mathcal{O}$ from transforms to correctly changed objects

1: **procedure** TRANSFORMS-SEARCH($\mathcal{G}_t, \mathcal{E}$)
2:     LVL, B, E $\leftarrow 0, \emptyset, \emptyset$      ▷ Initialize search state
3:     **while** LVL $\leq$ LIM **do**
4:         **for** $T \in$ NEW-TRANSFORMS($\mathcal{G}_t$, LVL, B) **do**      ▷ For all transforms with cost LVL
5:             EVAL $\leftarrow \{[\![T]\!](\omega_i) \mid \langle i, o \rangle \in \mathcal{E}, \omega_i \in i\}$      ▷ Apply transform on objects in input grids from $\mathcal{E}$
6:             **if** EVAL $\cap \bigcup_{\langle i, o \rangle \in \mathcal{E}} \{\omega_o \mid \omega_o \in o\} \neq \emptyset$ **then**      ▷ $T$ covers a subset of objects
7:                 $\mathcal{O}[T] \leftarrow$ EVAL      ▷ Store the transform and objects covered by it
8:             **else if** EVAL $\in$ E **then**
9:                 **continue**      ▷ $T$ is observationally equivalent to another transform in B
10:             **if** $\bigcup_{T \in \mathcal{O}} \mathcal{O}[T] = \bigcup_{\langle i, o \rangle \in \mathcal{E}} \{\omega_o \mid \omega_o \in o\}$ **then**      ▷ All objects are correctly transformed
11:                 **return** $\mathcal{O}$
12:             B[LVL] $\leftarrow$ B[LVL] $\cup \{T\}$      ▷ Add transform to the bank, indexed by cost for later search
13:             E $\leftarrow$ E $\cup$ EVAL      ▷ Cache evaluation result
14:         LVL $\leftarrow$ LVL $+ 1$
15:     **return** $\bot$      ▷ Cost limit reached

---

# K   The ARC Synthesis Algorithm

**Overall Synthesis Algorithm**   The overall synthesis algorithm takes as input a set of input-output grids $\mathcal{E}$, along with grammars $\mathcal{G}_t$ and $\mathcal{G}_f$. We sample candidate solutions from an LLM by constructing a prompt using $\mathcal{E}$. These solutions are used to initialize the weights of production rules in the transform and filter grammars, $\mathcal{G}_t$ and $\mathcal{G}_f$, respectively. We optimize the search by using a divide and conquer approach: first, a TRANSFORM-SEARCH procedure searches for transforms, mapping each to its correctly transformed objects in $\mathcal{O}$. Following this, a search for filters is initiated using the FILTER-SEARCH procedure. If a filter is found for each transform, the algorithm terminates and returns $\mathcal{M}$, which maps each transform to its corresponding filter. The algorithm described above terminates after the first solution is found, but we keep searching for a smaller set of transforms [6].

**Transform Search Algorithm**   The transform synthesis algorithm in Algorithm 3 takes as input a PCFG $\mathcal{G}_t$ and $\mathcal{E}$. It enumerates transforms in the order of increasing discrete costs according to $\mathcal{G}_t$.

The algorithm starts with the following initial state: 1) a cost level (LVL) equal to 0 in order to keep track of the current cost during enumeration, 2) a program bank (B) that indexes the enumerated transforms by their cost for efficient retrieval, and 3) an evaluation cache (E) that stores the result of all evaluated transforms within B. At each iteration, the algorithm explores the space of all new transforms generated by the NEW-TRANSFORMS procedure for the current cost level.

On line 5 in Algorithm 3, the enumerated transform $T$ is applied to each object in the input grids from $\mathcal{E}$. If $T$ correctly transforms a subset of objects, $T$ and the objects covered by it are stored in map $\mathcal{O}$ indexed by the transform (line 7). When the transforms in $\mathcal{O}$ cover all grid objects, $\mathcal{O}$ is returned (line 10-11). For transforms with objects bound by a filter, such as in `update_color(color_of(other))`, we consider all possible values (of color) that could be assigned and yield concrete transforms corresponding to each of those assignments.

**Filter Search Algorithm**    The filter search algorithm takes as input a filter PCFG $\mathcal{G}_f$, $\mathcal{E}$, and the map $\mathcal{O}$ returned by the transform search in Algorithm 3. The filter search proceeds in a similar manner as the transforms search wherein it enumerates filters in the order of increasing cost as per the PCFG $\mathcal{G}_f$. It initiates a new search to find a filter for each transform in $\mathcal{O}$. Each enumerated filter expression is evaluated on all objects in the input grids. If the objects for which the filter is True are the same as the objects covered by the transform, we have found a filter for this transform. Once a filter is found for each of the transforms in $\mathcal{O}$, we return the solution map $\mathcal{M}$.

