# OpenReview forum: "HYSYNTH: Context-Free LLM Approximation for Guiding Program Synthesis"
_NeurIPS.cc/2024/Conference — NeurIPS 2024 poster_

### Official Review · Reviewer_MWWa · 2024-07-12

**Soundness:** 4
**Presentation:** 4
**Contribution:** 3
**Rating:** 7
**Confidence:** 5

**Summary:**

This paper presents a hybrid program synthesis approach that strengthens the classic bottom-up search with an LLM-guided prior distribution, given as a probabilistic context-free grammar (PCFG). The focus of this paper is to solve complex PBE tasks where programs in unfamiliar DSL are difficult for LLMs to generate directly, and traditional combinatorial search is infeasible. The key insight is that LLM-predicted programs (although incorrect by themselves) can provide valuable intuition regarding which operator or component to use in the correct program. As such, HYSYNTH samples a set of full programs from LLM to train an approximation PCFG model. To perform the bottom-up synthesis, HYSYNTH uses an off-the-shelf or custom synthesizer for the specific domain.
The experimental results show its improvement against both vanilla LLM generators and non-LLM-based synthesizers.

**Strengths:**

- It proposes a promising hybrid approach that combines the efficiency of formal program synthesis and the power of LLMs.
- Compared to training new models for every new DSL/domain, the proposed approach only needs to extract a small PCFG from a few LLM samples, which is lightweight and robust.
- The approach is general; in particular it can be applied to various domains (reasoning about grid puzzles, tensor manipulations, and string manipulations) and with different LLMs. It can outperform baselines substantially according to the experiments.
- The problem is well-motivated, and the limitations and related work are discussed in an insightful and thorough manner.

**Weaknesses:**

- The datasets are small and domain specific, and requires implementing a synthesizer. It is unclear whether such an approach can scale to more complex programs, for example to program with loops, or even general-purpose programs.
- It mentions in the limitation section that sampling from LLM is costly, and it uses different models like GPT3.5, GPT-4, and DeepSeek, but did not provide a comparison of the costs.

**Questions:**

- Using PCFG as the approximation model seems too simple at first glance, but it works well in the evaluated benchmark. It is also surprising to see that only 10 samples can often achieve comparable performance as the full 100-sample approach (Appendix C). This leads me to wonder whether the model only captures very superficial patterns from the LLM samples, and whether an even simpler surrogate model could be used. For example, It is mentioned in line 151-157 that GPT-4o predicts the relevant components with high accuracy, and never uses irrelevant components in the example. If we directly remove such irrelevant components from the CFG and directly use the search algorithm, how would such a baseline perform?
- I am not entirely convinced by the superior performance of HYSYNTH over using LLMs purely without search. For example, for the ARC task, the HYSYNTH approach uses a divide-and-conquer strategy; it would be interesting to see how the LLM baseline performs when it also applies such a strategy. Also, in the paper ExeDec, Section 5.2, the experiments show that presenting DSL programs as Python functions can enable the LLM to better leverage its general knowledge from pretraining and improve the result. I wonder if such a prompting technique would improve the performance of pure LLMs.

[1] Shi, Kensen, et al. "ExeDec: Execution Decomposition for Compositional Generalization in Neural Program Synthesis." The Twelfth International Conference on Learning Representations. (ICLR 2024)

**Limitations:**

The limitations are well addressed.

---

> ### Author Rebuttal · Authors · 2024-08-05
>
> Thank you for your detailed review and thoughtful comments. Please note our top-level comment with additional experimental results. Below we address specific comments and questions.
>
> **Q: The datasets are small and domain specific, and requires implementing a synthesizer. Can HySynth scale to more complex programs, for example to program with loops, or even general-purpose programs?**
>
> It is true that we have not evaluated the approach on more complex programs with loops, but there is relevant work in the program synthesis community on extending bottom-up search to programs with local variables and loops [1,2].
>
> [1] LooPy: interactive program synthesis with control structures  https://dl.acm.org/doi/pdf/10.1145/3485530
>
> [2] Efficient Bottom-Up Synthesis for Programs with Local Variables https://dl.acm.org/doi/pdf/10.1145/3632894
>
> **Q: It mentions in the limitation section that sampling from LLM is costly, and it uses different models like GPT3.5, GPT-4, and DeepSeek, but did not provide a comparison of the costs.**
>
> The inference cost of the LLMs is dependent on the API pricing; with GPT3.5 and DeepSeek being cheaper than GPT-4o and the number of samples drawn per LLM are roughly the same. We are happy to include a cost analysis across LLMs in an updated version of the paper.
>
> **Q: If we remove irrelevant components not present in LLM solutions from the CFG  and use the search algorithm, how would such a baseline perform?**
>
> This is an interesting experiment and we have now performed it on the String and Tensor domains. More specifically, we evaluated an ablation, where the surrogate model is a “binary PCFG”, i.e. a CFG that only includes components mentioned by the LLM.
> You can see the graph of the results in the pdf attached to the Global Response. It performs worse than HySynth because the search might exclude essential components from the grammar. This experiment further highlights the balance achieved by HySynth in terms of prioritization based on LLM guidance.
>
> **Q: for the ARC task, HySynth uses a divide-and-conquer strategy; how would an LLM baseline that also applies such a strategy perform?**
>
>  If we understand correctly, the reviewer is suggesting to query the LLM for filters and transforms separately and then combine the results in some way. We argue, however, that such a  technique would not count as an LLM baseline, but rather as a different hybrid algorithm (only slightly less advanced than HySynth). Like HySynth, it would combine LLMs with a semantics-based pruning technique  originally proposed in the program synthesis community (in this case, divide-and-conquer synthesis [3]).
>
> [3] Scaling Enumerative Program Synthesis via Divide and Conquer
> https://www.cis.upenn.edu/~alur/Tacas17.pdf
>
> **Q: Will presenting DSL programs as Python functions enable the LLM to better leverage its general knowledge from pretraining and improve the results?**
>
> This is an interesting suggestion that could perhaps improve the performance of LLMs. However, we consider non trivial prompting and fine tuning techniques as being orthogonal to our contribution, which focuses on *using existing LLM solutions* more effectively.

---

> ### Comment · Reviewer_MWWa · 2024-08-09
>
> Thanks for the responses and adding the new experiments – I'll keep my score and raise the confidence as the new "binary PCFG" baseline addresses my concern and highlights the effectiveness of design components.

---

### Official Review · Reviewer_XXbQ · 2024-07-12

**Soundness:** 3
**Presentation:** 3
**Contribution:** 3
**Rating:** 6
**Confidence:** 3

**Summary:**

The paper introduces a new approach for solving structured prediction and reasoning tasks by leveraging the programming and planning capabilities of LLMs to enhance bottom-up search in program synthesis.

Specifically, HYSYNTH initially generates preliminary solutions directly from the LLM. Since direct LLM sampling performs poorly in unfamiliar DSLs, HYSYNTH does not use these samples directly but calculates the probabilities for a PCFG grammar from them, incorporating LLM guidance into the bottom-up search (CFG -> PCFG).

Experiments show that HYSYNTH consistently outperforms both the baseline synthesizers and ablations, and is effective even with a smaller number of samples.

**Strengths:**

The HYSYNTH method can be implemented directly on existing LLMs without additional model fine-tuning, allowing its performance to improve as the models advance.

HYSYNTH performs well with fewer samples, consistently surpassing current methods, making it robust and efficient.

Experiments demonstrate that HYSYNTH consistently outperforms both baseline synthesizers and ablations across various configurations.

**Weaknesses:**

The use of LLMs in HYSYNTH appears simplistic and seems not to fully leverage the capabilities of LLMs.

HYSYNTH's performance depends on the LLM's capabilities, including additional inference time and prediction accuracy.

**Questions:**

The experiments show that HYSYNTH solves more problems within the same time limit. Does the time reported include the time taken for sampling from the LLM and training the surrogate model? What percentage of the total time does LLM inference take?

For GPT-4o's 0% performance on the percentage of syntactically valid completions in the STRING domain, can a more detailed analysis be provided? Could this be improved with a simple CoT strategy or other methods?

**Limitations:**

The authors have addressed the limitations of their work.

---

> ### Author Rebuttal · Authors · 2024-08-06
>
> Thank you for your detailed review and thoughtful comments. Please note our top-level comment with additional experimental results.
>
> **Q: The experiments show that HySynth solves more problems within the same time limit. Does the time reported include the time taken for sampling from the LLM and training the surrogate model? What percentage of the total time does LLM inference take?**
>
> See Global Response Q1.
>
> **Q: For GPT-4o's 0% performance on the percentage of syntactically valid completions in the STRING domain, can a more detailed analysis be provided? Could this be improved with a simple CoT strategy or other methods?**
>
> See Global Response Q2.

---

### Official Review · Reviewer_FZiE · 2024-07-12

**Soundness:** 3
**Presentation:** 3
**Contribution:** 3
**Rating:** 7
**Confidence:** 4

**Summary:**

This paper introduces HySynth, an approach to program synthesis that combines (1) sampling programs from a large language model (LLM) to train a probabilistic context free grammar (PCFG) with (2) applying bottom-up enumerative search guided by the PCFG to solve programming by example tasks. It applies this approach to three domains: the abstract reasoning corpus (ARC), tensorflow expression synthesis, and string expressions (SyGuS). By learning the weights of the PCFG from the samples drawn from the LLM, HySynth produces state of the art results on all three domains, solving more problems in less wall-clock time. The main contribution is the use of an LLM to choose weights to guide a bottom-up enumerative program synthesis search system.

**Strengths:**

The paper is the first (concurrent with Li et al [29]) to use pretrained LLMs to guide search-based program synthesis. The approach introduced by HySynth -- to sample programs from a pretrained LLM to learn the weights for a PCFG, and then to guide bottom-up enumerative search according to these weights -- is a natural and simple idea, and the paper demonstrates that it works well across a variety of domains. The simplicity of the approach is a strength, making it possible to apply to a range of domains -- wherever a CFG can describe the DSL used in the domain. The simplicity and generality of the approach make the contribution significant: it is quite likely that additional methods that learn to perform a fast search guided by an LLM will follow in future research, as program synthesis with LLMs is an important and growing area.

The experimental results are robust, showcasing the HySynth method across a good diversity of domains and yielding clear improvements in each. (See also Weakness 1 however.)

The paper is clearly written, including a robust and clear background section, a clear statement of the method and experimental results, and well contextualized in the context of the literature.

**Weaknesses:**

For the direct sampling baseline, there is either a clarity or methodological issue. It is unclear how many samples are drawn from the LLM in the direct sampling approach, and if the number of samples is more than 1, it is not clear what is done to produce diversity (i.e. setting the temperature during sampling, or sampling without replacement e.g. with unique randomizer [1]). From the horizontal lines in Figure 4a, 4b, and 4c, it seems overwhelmingly likely that only a single sample is drawn from GPT4o for the direct sampling baseline, which places this baseline at a disadvantage since >>1 samples can be drawn within 10 minutes. The disadvantage is compounded because direct sampling is trivially parallelizable, so the number of samples that could be drawn in this time is large. And as one further point in favor of drawing multiple samples for the direct sampling baseline, the HySynth approach itself uses as many as N = 100 samples from the LLM (line 273). I recognize this may be a costly baseline to run (I have not computed the cost), and suggest a reduced time limit if the cost is prohibitive for the 10m time limit.

[1] Incremental Sampling Without Replacement for Sequence Models https://arxiv.org/pdf/2002.09067

The other weaknesses I observed are touched upon in the limitations section of the paper. The main limitation of the significance of the work is (as stated in the paper) that is requires a DSL, and particularly one that has a CFG, and so a custom synthesizer for each DSL, for the method to be applied. This requires meaningful work to apply HySynth to a new domain.
Another concern (also addressed in the limitations section) is the possibility of data leakage in the evaluations. I share the authors' view that this is not a major issue _provided_ they address the first weakness I state above for the direct sampling baseline.

**Questions:**

Is the time to draw N samples from the LLM and construct the PCFG included in the plots in Figure 4a, 4b, 4c? If not, I would encourage doing so.

Why do you include few-shot examples when prompting for the tensor domain, but not for the string domain (or have I misread this?)? It seems more important for the string domain given that 0% of the LLM-generated programs in the string domain are valid completions.

nit: Figure 2 suggests the I/O examples are provided as an input to the synthesizer, rather than just the PCFG. I don't believe this is correct or intended.

Is it correct to state that the PCFG is reasonably approximating the LLM's condition distribution over output programs for a given prompt (line 65)? Though it is trained to approximate this, I expect it is quite a poor approximation. And then the approach does not sample from the PCFG, but rather enumerates from it, making it unclear to me that a good unbiased approximation of the LLM's conditional distribution is what is desired (certainly if the PCFG could perfectly mimic this distribution that would be great!, but perhaps there is better; beyond wanting to somewhat approximate the LLM's condition distribution, increased functional diversity over the 10m search period is also important.)

**Limitations:**

The limitations section (4.3) captures the main limitations of the system well and articulates them clearly. The Broader Research Impacts appendix (K) touches on the research, albeit not societal, impact of the work. This seems sufficient to me.

---

> ### Author Rebuttal · Authors · 2024-08-06
>
> Thank you for your detailed review and thoughtful comments. Please note our top-level comment with additional experimental results. Below we address specific comments and questions.
>
> **Q: For the direct sampling baseline, clarification about how many samples are drawn from the LLM in the direct sampling approach, and if the number of samples is more than 1, what is done to produce diversity?**
>
> There has been a misunderstanding: the results of the direct sampling approach are based on 100 samples (see line 251), the same number that is used to learn the PCFG. We also report the temperature and other prompting settings in appendix J – we sample with a high temperature to get diverse solutions. Please see Global Response Q1 for why we decided to separate LLM sampling time from symbolic synthesis time.
>
> **Q: The main limitation of the work is that is requires a DSL, and particularly one that has a CFG, and so a custom synthesizer for each DSL, for the method to be applied. Can this be overcome?**
>
> Our approach does require a CFG for the DSL but a custom algorithm is optional. In this paper, for the String domain we use a generic synthesizer that takes a grammar (and an interpreter) for the DSL as an input.
>
> **Q: Is the time to draw N samples from the LLM and construct the PCFG included in the plots in Figure 4a, 4b, 4c?**
>
> See Global Response Q1.
>
> **Q: Why do you include few-shot examples when prompting for the tensor domain, but not for the string domain? It seems more important for the string domain given that 0% of the LLM-generated programs in the string domain are valid completions.**
>
> See Global Response Q2.
>
> **Q: Why are I/O examples provided as an input to the synthesizer, rather than just the PCFG?**
>
> The I/O examples need to be provided as input to the synthesizer in order to test the correctness of generated programs and prune the search space based on observational equivalence (see lines 5, 6, and 8 in Algorithm 1).
>
> **Q: Is it correct to state that the PCFG is reasonably approximating the LLM's condition distribution over output programs for a given prompt?**
>
> You are right that a PCFG is still a poor approximation (even for the conditional distribution). What we meant to say here is that, for a specific task, a PCFG is able to capture just enough signal from the LLM that it can guide enumerative search. We’re happy to update the phrasing!

---

> > ### Comment · Reviewer_FZiE · 2024-08-13
> > **Response to Rebuttal**
> >
> > Thank you for your rebuttal, and in particular Global Responses 1 and 2.
> >
> > Regarding Global Response Q1 (Q1, Q3 in "Rebuttal by Authors"):
> >
> > > The time to draw those samples is excluded from the graphs in Fig 4.
> >
> > Thank you for clarifying this. This is a very important detail to have omitted, and it does meaningfully change the results of the paper / what is conveyed in Figure 4. The baselines ARGA, Probe, TF-Coder, and Unguided are being given a 268-285 second disadvantage in Figure 4 compared with HySynth and No-Search. Eyeballing it, it looks like HySynth still outperforms the baselines even when this adjustment is accounted for, but not as immediately or decisively as Figure 4 currently conveys.
> >
> > My recommendation for making this clear would be to include the sampling time in the Figure 4 plots, giving all methods the same total time (not just the same search time) for a fair comparison. Making clear the total time spent by each method (not just the time spent searching) is necessary for a fair comparison of the methods if you wish to claim that HySynth outperforms the other methods (a key claim of the paper, line 86).
> >
> > > That said, we did compute the time it took to sample 100 solutions, and it amounts to 285 secs for tensor domain and 268 secs for the string domain.
> >
> > Thanks for measuring this. This suggests that running the No-Search / "GPT-4o" baseline for 10 minutes would involve drawing no more than 225 samples, and so the cost might be quite affordable. A back-of-the-envelope calculation suggests the total cost would be between $3 - $4 USD per domain (so up to $12 USD total). If you do this, the Fig. 4 plot for the "GPT-4o" baseline would then show progress over time like the other methods, rather than being a fixed horizontal line.
> >
> > > 1) LLM sampling incurs not only time cost, but non-trivial monetary cost, while symbolic search is virtually free
> >
> > Yes, it's true that sampling is more expensive than search per unit time, but I don't think this justifies ignoring the sampling time. (e.g. the caption for Fig 4 currently reads as "Number of benchmarks solved by HYSYNTH as a function of time", not even specifying search time; similarly the language used in 4.2 is "We compare the time to solution for the main HYSYNTH configuration, baseline synthesizers, and the two ablations; the results for the three domains are shown in Fig. 4", again giving the impression that this measures total time.)
> >
> > Regarding Global Response Q2 (Q4 in "Rebuttal by Authors"):
> >
> > > Providing in-context examples is unlikely to help because these examples would have a different grammar, and only confuse the model.
> >
> > This is possible, but I would not be surprised if in-context examples did help, even if they came from different grammars. (The way I expect you would prompt the model, the grammars would be included in the in-context examples.)
> >
> > --------
> >
> >
> > Q2: Thank you for clarifying.
> >
> > Q5: Thank you for clarifying; that was my mistake.
> >
> > Q6: Thank you for your comment and agreeing to update the text.

---

> > > ### Author Response · Authors · 2024-08-14
> > >
> > > Non-LLM baselines (ARGA, Probe, TF-Coder, and Unguided) are at a disadvantage:
> > >
> > > > Sorry, our explanation was confusing: the time reported in the response is for the entire dataset; it only takes ~4 seconds to sample 100 solutions for a single problem from gpt4o. This makes the disadvantage quite small, and our claim that HySynth outperforms all the non-LLM baselines would definitely still hold if we added this sampling time to HySynth’s results. We will make a plot with the sampling time incorporated in future revisions in order to convey the full picture.
> > >
> > > LLM baseline is at a disadvantage because we did not sample until timeout:
> > >
> > > > In light of the last paragraph, sampling for 10 min **per problem** would actually be quite expensive (but we can perform this experiment with a shorter timeout, if the reviewer thinks it’s necessary). In general, we still believe that cost  is a major limiting factor for the LLM baseline, rather than just time (especially since, as you mentioned, samples can be generated in parallel). So, to put all techniques on equal footing, it makes more sense to limit both time and cost for all the techniques. In our paper we use sample size (100) as an approximation of the cost limit, but we could also conduct experiments with an actual cost limit, if the reviewer finds this necessary (in which case, the sample size will be different for different problems and LLMs).

---

> > > > ### Comment · Reviewer_FZiE · 2024-08-14
> > > >
> > > > Thank you for clarifying. The disadvantage is much smaller than I thought in writing my previous response.

---

### Author Rebuttal · Authors · 2024-08-06

We thank the reviewers for their time, valuable comments, and encouraging feedback! In the global part of the response, we answer shared questions and provide a list of changes we plan to make in a revised version of the paper.

**Q1 (Reviewers FZiE and XXbQ): Does the time reported in Fig 4 include the time taken for sampling from the LLM and training the surrogate model? What percentage of the total time does LLM inference take? The LLM baseline is at a disadvantage because many more samples could be drawn within the timeout of 10 minutes.**

In our experiments, the results for both the LLM baseline and HySynth are based on 100 samples from the LLM. The time to draw those samples is excluded from the graphs in Fig 4.
Reviewer FZiE is right that many more samples could be drawn within 10 minutes. Indeed, one way to conduct our experiments would be to give all techniques the same amount of total time, which can be used towards either LLM sampling or symbolic synthesis. We thought such setup would not be fair for the following two reasons: 1) LLM sampling incurs not only time cost, but non-trivial monetary cost, while symbolic search is virtually free; 2) the time cost of LLM sampling depends on many incidental factors out of our control (e.g. network latency, server load). We chose instead to completely separate the sampling stage from the synthesis stage and to only report the synthesis times in the figures (which is why the LLM-only line is always horizontal, as the reviewer notes).
That said, we did compute the time it took to sample 100 solutions, and it amounts to 285 secs for tensor domain and 268 secs for the string domain. Learning the PCFG takes less than 1 sec for all domains. We will report these times in the next version of the paper.

**Q2 (Reviewers FZiE and XXbQ): Why does GPT-4o have 0% correctness and syntactic validity on the String domain? Could this be improved with more advanced prompting techniques? Why do we not provide in-context examples for this domain, like for Tensor?**

This is due to the specifics of the SyGuS benchmark, from which our String problems are drawn. In this benchmark, each problem comes with a custom restricted grammar; all grammars are subsets of the full SyGus grammar but purposefully impose additional restrictions on the solution (for example, the synthesizer is usually allowed only a handful of string constants, which excludes trivial solutions that simply reproduce the given output examples). In the experiments reported in the paper, we judge syntactic validity against these custom grammars (which is how the SyGus competition is also judged), and the LLMs turn out to be quite poor at following these syntactic restrictions, even though the grammar is given in the prompt.
Providing in-context examples is unlikely to help because these examples would have a different grammar, and only confuse the model.
If we were to relax the definition of syntactic validity, ignoring the custom grammars and allowing the full SyGuS grammar, LLM solutions would achieve a syntactic validity of 60.5% and solve 20/70 problems correctly (which is better than Unguided search but worse than Probe and HySynth.). We will include these additional results in the paper along with a detailed analysis.

**Change list**
1. We will report the time and monetary cost of sampling from the different LLMs in our experiments (as mentioned in Q1 above).
2. We will clarify why LLMs get 0% syntactic validity on the String domain and add the results of our new experiments with a relaxed notion of validity (as explained in Q2 above).
3. We will include the results of an ablation study using a simpler surrogate model, which simply excludes the components not used by the LLM (as suggested by Rev MWWa; preliminary results reported in the attached PDF).

---

### Decision · Program_Chairs · 2024-09-25

**Decision:**

Accept (poster)

**Comment:**

This paper proposes performing bottom-up program synthesis search but using a PCFG derived from LLM-generated samples to guide the search, leading to improved performance in a variety of synthesis domains/datasets.

I recommend to **accept** this paper. Reviewers agreed that the approach is well-motivated and general, the experiments are convincing and thorough, and the writing is clear. The authors answered reviewers' questions and resolved their concerns in the rebuttal. All reviewers are in agreement about acceptance.

I am confident that this paper meets the bar for acceptance. However, I was on the fence about recommending poster or spotlight. On the positive side, the reviewers had a high average score (6, 7, 7) and I agree that the paper and underlying research were done very well. But after much thought, I am recommending a poster presentation because I feel the core idea is very simple and natural, which is a good thing when trying to replicate the work, but it also means the work is a bit more incremental than I would imagine for a spotlight paper. The authors cite multiple prior works which learn how to guide combinatorial program synthesis search by training a model on data in a particular domain, and this paper's contribution is using an LLM to provide this guidance instead, more generally (across multiple domains) and without training. This feels like an instance of a common pattern of generalist LLMs being used to replace specialist models trained from scratch. Under this lens, I am not so surprised that the approach works. Again, I still think the work was done very well and it is a valuable contribution to the AI-for-code research community to have solid confirmation that this approach does indeed work, but I would expect a spotlight paper to provide a bit more novelty.